# Evaluation of Clastogenic and Aneugenic Action of Two Bio-Insecticides Using Allium Bioassay

**DOI:** 10.3390/jox15020035

**Published:** 2025-02-27

**Authors:** Georgiana Duta-Cornescu, Maria Liliana Dugala, Nicoleta Constantin, Maria-Daniela Pojoga, Alexandra Simon-Gruita

**Affiliations:** Genetics Department, Faculty of Biology, University of Bucharest, Intrarea Portocalelor 1-3, Sector 6, 060101 Bucharest, Romania; dugala.maria-liliana22@s.bio.unibuc.ro (M.L.D.); nicoleta.constantin@bio.unibuc.ro (N.C.); daniela.pojoga@bio.unibuc.ro (M.-D.P.); alexandra.simon@bio.unibuc.ro (A.S.-G.)

**Keywords:** bio-pesticides, *A. cepa* assay, cytogenotoxicity, phytotoxicity, bio-pesticides

## Abstract

It is well known that modern agriculture would not be able to meet the current demand for food without the help of pesticides. However, conventional pesticides have been proven to be extremely harmful to the environment, to the species they are applied to, and, ultimately, to humans. As a result, bio-pesticides have been introduced in recent years and include natural substances that control pests, such as biochemical pesticides, microorganisms used as pest control agents (microbial pesticides), and pesticide substances produced by plants containing added genetic material, known as plant-incorporated protectants (PIPs). Although these are natural products, their widespread use has led to an increased presence in the environment, raising concerns regarding their potential impact on both the environment and human health. The aim of our study was to determine the phyto- and cytogenotoxicity caused by two insecticides, both certified for use in ecological agriculture: one biochemical (BCP) and the other microbial (MP), which were applied in three concentrations (the maximum recommended concentration by the manufacturers (MRFC), 1.5X MRFC, and 2X MRFC) to the meristematic root tissues of *Allium cepa*. The results were compared to a negative control (tap water) and a positive control (a chemical pesticide (CP) containing mainly Deltamethrin). Phytotoxic and cytogenotoxic effects were analyzed at two time intervals (24 and 48 h) by measuring root length, growth percentage, root growth inhibition percentage (phytotoxicity tests), and micronuclei frequency and chromosome aberrations (anaphase bridges, chromosomal fragments, anaphase delays, sticky chromosomes, laggard/vagrant chromosomes) (cytogenotoxicity analyses), respectively. The tests conducted in this study showed that the microbial insecticide provides greater safety when applied, even at higher doses than those recommended by the manufacturers, compared with the biochemical insecticide, whose effects are similar to those induced by the chemical pesticide containing Deltamethrin. However, the results suggest that both insecticides have clastogenic and aneugenic effects, highlighting the need for prior testing of any type of pesticide before large-scale use, especially since the results of the *A. cepa* tests showed high sensitivity and good correlation when compared to other test systems, e.g., mammals.

## 1. Introduction

According to the HSE (Health and Safety Executive—Britain’s national regulator for workplace health and safety), pesticides, also known as ‘plant protection products’ (PPP), are used to control pests, weeds, and diseases, being divided into insecticides, fungicides, herbicides, molluscicides, and plant growth regulators. They can exist in many forms, such as solid granules, powders, or liquids, and consist of one or more active substances co-formulated with other materials [1].

Pesticides are indispensable in modern agricultural production because they allow to obtain increased quantities of products. It is estimated that without the use of pesticides, there would be a 78% loss in fruit production, a 54% loss in vegetable production, and a 32% loss in grain production [2]. According to Carvalho, 2017, pesticide production increased at a rate of about 11% per year, from 0.2 million tons in the 1950s to more than 5 million tons by 2000 [3], with over three billion kilograms of pesticides used worldwide every year [4]. However, only 1% of total pesticides are effectively used to control insect pests on target plants [5], most of the amount used remaining in the environment or acting on non-targeted species [6,7]. Therefore, the indiscriminate use of pesticides has led to serious health and environmental risks [8,9,10,11]. Due to the increased concerns regarding the environmental and health effects of pesticides, intensive research efforts were put into understanding farmers’ pesticide use behavior and also in developing new types of pesticides, with a minimum to no effect on the environment or non-targeted species. Currently, alternative solutions are being considered and sought after. One of the sustainable solutions is the use of bio-pesticides, as they can be less toxic than chemical pesticides, can be highly specific to the target pest, break down quickly, and have less chance of developing resistance [12].

According to the EPA (United States Environmental Protection Agency), bio-pesticides include naturally occurring substances that control pests (biochemical pesticides), microorganisms that control pests (microbial pesticides), and pesticide substances produced by plants containing added genetic material (plant-incorporated protectants) or PIPs [13]. While conventional pesticides are usually artificial synthesized chemicals that directly kill pest, the bio-chemical pesticides are naturally occurring substances that can be used to control pests through non-toxic mechanisms. Microbial pesticides consist of microorganisms (e.g., a bacterium, fungus, virus or protozoan) as the active ingredient and can control many kinds of pests, although each separate active ingredient is relatively specific for a target pest [14,15].

Even with the emergence of these new bio-pesticides, in the intensive agriculture, the continuous increase in requirements for agricultural products leads to an inappropriate, often indiscriminate, use of pesticides and fertilizers. Therefore, the lack of adequate management and studies to monitor the impact of these products both on the environment and on plants themselves can have a negative impact on human health as well [16,17,18].

Some of the most used methods for evaluating the toxic effect of pesticides, including insecticides, are phyto- and cytogenotoxicity assays [19,20]

Usually, phytotoxicity tests are used for assessing the impact of various compounds on seed germination and subsequent growth, while plant cytogenotoxicity tests evaluate the impact of a chemical compound on the cell cycle, chemicals that can induce different types of chromosome aberrations (CA). CA represents modifications in either chromosomal structure (due to DNA breaks, inhibition of DNA synthesis and replication of altered DNA) or in the total number of chromosomes (e.g., aneuploidy and polyploidy, as consequences of abnormal segregation of chromosomes) [21,22]. There is a large amount of research indicating that analysis of the different chromosome aberration types in all phases of the cell cycle permits a more comprehensive and accurate evaluation of the clastogenic and/or aneugenic effects of the tested agents [21,23,24,25,26] and points that modifications like chromosome bridges and breaks are indicators of a clastogenic action, whereas chromosome losses, delays, adherence, and multipolarity result from aneugenic effects [27,28].

The *A. cepa* bioassay has been used since 1930 to detect the effect of different chemical compounds on the plant genome and is still widely applied today because it is simple, reliable, cheap, and accessible [29,30,31]. Moreover, many studies have identified similarities between the negative effects (e.g., causing CAs) of some pesticides on cellular DNA from *A. cepa* and from various mammalian species, including humans [29,30,31]. For example, Fiskesjö [32], when studying the effect of a number of compounds (salt solutions, industrial wastewaters, etc.), showed that the sensitivity of the Allium test was on the same level as the test on human lymphocytes, while Rank and Nielsen [33] indicated an 82% correlation between *A. cepa* tests and carcinogenicity tests in rodents when testing the effect of five mutagenic or carcinogenic chemicals usually found in wastewater. In another study regarding the cytogenetic effect of pyrethroid insecticides, Chauhan et al. [30] reported a good correlation between the *A. cepa* test system and those on mammals (rodents’ bone marrow and human lymphocytes), suggesting that the *A. cepa* test can be validated as an alternative to the mammal test systems for monitoring the genotoxic potential of environmental chemicals, such as pesticides.

*A. cepa* has a small genome (2n = 16) and large chromosomes, which makes it possible to observe their behavior during mitotic divisions [31,34] and allows for the evaluation of different genetic effects, as well as understanding the action mechanisms of various categories of chemical compounds [21].

In this research, the *A. cepa* bioassay was used to assess the phytotoxic and cytogenotoxic impact of two bio-insecticides administered in different concentrations (the maximum dose recommended by the manufacturers and higher). The two tested insecticides belong to two different categories: one is included in the “bio-chemical pesticides” class (BCP) and contains Spinosad, Propanediol, and 1,2 Benzothiazoline, and the other one belongs to the “microbial pesticides class” (MP) and contains *Bacillus thuringiensis*, *Lysinbacillus sphaericus*, *Bacillus subtilis* and other components), certified to be used in ecological agriculture.

The use of positive control (PC), i.e., a compound known to cause various cytotoxic and genotoxic changes, is essential [19,35,36]. Many of the insecticides used wordwide contain Delthametrin [37] and, according to the IOBC (International Organization of Biological Control) rating, Deltamethrin is placed in category 2 level of toxicity, which can be interpreted as moderately harmful [38]. Therefore, Deltamethrin was used as a positive control because, in numerous studies, it proved to be toxic to plants and animals [39,40,41,42,43,44,45]. The potentially harmful effects of improper use of these pesticides on the environment and humans can thus be established.

## 2. Materials and Methods

### 2.1. Plant Material

Approximately equal sized (~2–2.5 cm diameter) and healthy onion bulbs (*Allium cepa* L., 2n = 16) were purchased at a local vendor and then carefully cleaned in order not to destroy the root primordia. The experiments were performed in triplicate for each treatment applied to onion bulbs.

### 2.2. Insecticide Solutions

Two insecticides from distinct categories of pesticides were tested: (1) a biochemical pesticide (BCP) that contains Spinosad (22.8%), Propanediol (≥3.0–<10.0%), polyalkylene glycol (≥3.0–<10.0%), and 1,2 Benzothiazoline (<0.05%) [46] and (2) a microbial pesticide (MP) containing super concentrated extract of feeding inhibitors, *Bacillus thuringiensis* 1 × 10^8^ CFU, *Lysinibacillus sphaericus* 1 × 10^8^ CFU, *Bacillus subtilis* 1 × 10^8^ CFU, and other components (20% mineral oil USP, 30% super concentrated vegetable extract, 50% glycerol, 0.5% red dye). Both insecticides are certified to be used in ecological agriculture, being considered effective against different species from Coleoptera, Lepidoptera, Thysanoptera, and Diptera orders that attack fruit trees, vegetables, and grape crops.

The positive control was represented by a chemical pesticide (PC) containing mainly Deltamethrin (10.5%) but also 2-Methylpropanol (1.00–3.00%), 2,6-Di-tert-butyl-4-methylphenol (0.10–0.25%), cyclohexanone (1.00–40.00%), aromatic hydrocarbons (20.00%), Benzenesulfonic acid, mono-C11-13 branched alkyl derivates, and calcium salts (1.00–5.00%).

In this assay, the three concentrations used were the maximum dose recommended by the manufacturers (MRFC), 1.5X MRFC and 2X MRFC. The experimental design and concentrations are presented in Table 1.

As negative control (NC), healthy onion bulbs grown only in tap water throughout the experiment were used.

For the preparation of the solutions for each of the three insecticides used (MP, BCP, and PC), the manufacturer’s instructions were followed. For example, in the case of MP, a concentration of 2% is recommended (i.e., 100 mL of insecticide solution diluted in 5 L of water). This was considered the MRFC (1X) experimental variant. The subsequent experimental variants (1.5 MRCF and 2 MRCF) were obtained by further adding insecticide solution in the same water volume (5 L), according to the desired concentrations. The same procedure was applied to all the insecticides. Since each experimental variant was performed in triplicate, the initial volume of the solution was 5 L to avoid any potential variation in the concentration of the solutions.

### 2.3. Phytotoxicity Assay

Prior to initiating the test, the outer scales of the bulbs and the dry bottom plate were removed without destroying the root primordia and germinated for 24 h in tap water, until the root primordia were visible. Then, the onion bulbs were transferred into the tested insecticide solutions. The negative control was further incubated in tap water.

The bulbs were allowed to sprout and grow for another 24 h and 48 h at room temperature (25 ± 2 °C), respectively. Numbers of sprouted roots were counted after each time interval and the root lengths were measured with the help of a calibrated ruler.

After each time interval, the length of the emerged roots was measured, and growth percentage and root growth inhibition percentage were computed using the following formulas:(1)$$Growth\;\%=\frac{Average\;root\;length\;in\;control\;plants}{Average\;root\;length\;in\;treated\;plants}\times 100$$(2)Root growth inhibition=Growth % in control plants−Growth % in treated plants

### 2.4. Cytogenotoxicity Assay

Six root tips, about 0.5 cm long, were randomly removed from each of the three replicates/treatment/time intervals and fixed in 3:1 absolute ethanol/glacial acetic acid for 24 h at 4 °C. The Feulgen staining, followed by the squash method, was applied to make the cytogenetic slides. The method involves hydrolysis with 1 N Hydrochloric acid (HCl) at a temperature of 60 °C and staining with basic fuchsin. After this treatment, the chromatin of the chromosomes stains red-violet, while the ribonucleic acid in the nucleoli and cytoplasm does not stain. The best slides in terms of biological material mounting, staining, and cellular integrity were selected for further analysis, which involved examining 500 cells/microscope slides.

The microscopic slides were visualized under the optical microscope with 10X, 20X, and 60X objectives. The photography was performed digitally with the Dino-Eye Eyepiece Camera (Dino-Lite Europe, IDCP B.V, Almere, The Netherlands) using the DinoCapture 2.0 (Windows) program.

For the cytogenotoxicity assay, for both time treatments (24 h and 48 h), mitotic index (MI), the % of cells in each mitotic phase were analyzed. MI was calculated using Formula (3).(3)$$MI=\frac{Number\;of\;dividing\;cells}{Total\;number\;of\;cells}\times 100$$

Then, micronuclei (MN) frequency and the number of chromosome aberrations (anaphase bridges, chromosomal fragments, anaphase delays, sticky chromosomes, laggard/vagrant chromosomes) were determined.

### 2.5. Statistical Analysis

All samples and controls were prepared in triplicate. The results are represented as the mean values ± their respective standard deviations. The phytotoxic and cytogenotoxic effects of different insecticide concentrations on *A. cepa* root cells were compared using a two-way analysis of variance (ANOVA). Means were compared with a *t*-test (*p* < 0.05), and the Bonferroni post-tests were applied to identify the statistical differences between the various test samples and the control. The Bonferroni test was applied because it is a statistical test used to reduce the instance of a false positive, being designed, in particular, to prevent data from incorrectly appearing to be statistically significant [47,48,49,50]. All statistical analyses were performed using the Data Analysis Tools in Microsoft Excel 2016. Differences between corresponding controls and exposure treatments were considered statistically significant at *p* < 0.05.

## 3. Results and Discussion

### 3.1. Root Germination and Growth Inhibition

The present study was carried out to determine the effect of different concentrations of two commercially available insecticides certified to be used in ecological agriculture, on germination and root elongation of *A. cepa* bulbs. The three tested concentrations were chosen considering the fact that today many farmers use phytosanitary products and especially insecticides in excess, often applying treatments with much higher concentrations than those recommended. This situation can be explained by the fact that using the maximum recommended doses or above can avoid or reduce the spread of insecticide-resistant insects [51].

As is shown in Table 2, the root length is negatively correlated with insecticide concentration and treatment time. For example, for BCP-treated samples, a significant reduction in the root’s length was observed at each of the three tested concentrations. The mean value of this parameter, after 48 h of treatment, was 0.75 ± 0.1, 0.82 ± 0.2, and 0.57 ± 0.2 compared with the negative control (NC) (2.75 ± 1.5). The level of root growth inhibition was comparable between the plants treated with Deltamethrin (PC) and the ones treated with BCP after 48 h of treatment (Table 2) and were correlated to the **root number**: at 48 h, the number of roots for PC plants was 30.5 ± 1.5 (at 1X experimental variant), 35 ± 7.7 (at 1.5X), and 31 ± 4 (at 2X), while for the BCP treated bulbs the values were even lower (28.5 ± 4.9; 19.5 ± 9.1, and 13.5 ± 0.7, respectively). These results indicate that the pesticides understudy were phytotoxic and that the level of this effect is concentration dependent.

The lowest level of root growth inhibition was recorded for MP, with the values ranging from 10.91% (1X variant—48 h of treatment) to a maximum of 22.73% (1.5X variant–after 48 h of treatment), values that are also correlated with the number of roots. These findings indicated that MP insecticide might have a very low to no phytotoxic effect, even when applied at a higher concentration than the one recommended by the manufacturer. Very intriguing observations were made for the 2X experimental variant, for which, at both time intervals (24 h and 48 h), higher values of root length and no. of roots were registered and, therefore, the calculated % of root growth inhibition had negative values (−4.45% at 24 h and −20% after 48 h). These results are consistent with the length and, respectively, the number of roots after 24 h and 48 h of treatment with MP 2X. Thus, after 24 h, the root length in 2X MP insecticide was 2.37 ± 0.4 cm vs. 2.2 ± 1.8 cm in NC, and after 48 h of treatment, the difference was even greater, 3.3 ± 0.2 cm vs. 2.75 ± 1.5 cm. Root number was also higher for 2X MP: 33.5 ± 0.7 after 24 h and 37.5 ± 0.1 after 48 h treatment compared with 30.5 ± 2.4 and 37 ± 1 in NC. These results might be explained by the presence, in larger quantities, of the bacteria contained in MP. Many phytostimulants are known to work through growth hormones (e.g., auxins that promote root growth, and gibberellins that cause stem elongation and flowering) and help regulate these hormones to improve plant growth [52]. Also, certain beneficial microorganisms, like plant-growth-promoting rhizobacteria (PGPR), stimulate plant growth by producing growth-promoting substances or improving the plant’s resistance to disease [53]. MP contains three microbial strains (*Bacillus thuringiensis*, *Lysinibacillus sphaericus* and *Bacillus subtilis*). There are a number of scientific papers that assessed the phytostimulation effect of *B. thuringiensis* [54,55]. Jensen et al. [56] showed that some *B. subtilis* strains can promote, in vitro, the increase of shoot and root surface area in *A. thaliana*, while Martínez et. Dussan [57] showed that *L. sphaericus* could be a key organism in some formula of biofertilizers, as it shows potential in the phytoremediation processes, in crop plant nutrition, and in growth in low nutrient and polluted soils.

In this context, it could be considered that MP had a phytostimulatory effect on root growth in *A. cepa* when it was applied ina 2X MRFC concentration.

### 3.2. Chromosomal Aberrations and Micronuclei Induction

The cytotoxic potential of new-generation insecticides on *A. cepa* root tip cells was investigated using microscopic analysis in order to observe disturbances on mitotic division and chromosomal aberrations. The analyses focused on cytogenetic changes both in interphase and during all stages of the mitotic division: prophase, metaphase, anaphase, and telophase.

For the four stages of the mitotic division, there was a greater variation in the percentage values of the dividing cells under the action of insecticides (BCP, MP, and PC) compared with NC, in which no significant differences were observed regardless of the duration of the experiment (Table 3).

The MI (mitotic index) of root meristematic cells decreased as insecticide concentration and exposure time increased in almost all treatments compared with the negative control (NC). These results proved the alteration exerted by insecticides on the growth and development of the meristematic cells of the root of onion bulbs, MI being an important parameter in monitoring the effect of different chemical substances on the environment and an appropriate indicator of the cytotoxicity level [21]. The highest MI values were observed in plants of NC, reaching 33.75% at 24 h and 45.25% at 48 h, which indicated normal mitotic activity. On the other hand, differences regarding the MI value varied depending on the type of insecticide used and its concentration. For example, in *A. cepa* bulbs treated with BCP, all concentrations tested (except the 0.04%, 1X, after 24 h) generated statistically significant MI decreases when compared to the negative control (Table 3). The lower MI (16.25%) was obtained for 1X experimental variant after 48 h of exposure, while for the other treatments, the values varied between 18.8% (2X after 24 h) and 21.38% (2X after 48 h). These results suggest that BCP insecticide might have an inhibitory effect on cell division.

MP treatment had a statistically significant inhibitory effect on *A. cepa* only at the 3% concentration (1.5X): MI after 24 h was 24.75 and 27.5 after 48 h compared with 33.75 at 24 h and, respectively, 45.24 at 48 h in NC (Table 3). For the 2X experimental variant (4% MP concentration), the registered MI index was lower than in controls but not statistically significant.

This effect can be explained by the fact that the insecticide has a clastogenic effect (because clastogenic agents can cause a reduction in MI [58] due to chromosomal fragmentation) but also aneugenic (because aneugenic agents, having cell division disturbing properties, can temporarily increase MI before causing cycle arrest or apoptosis [59,60]).

In all treatment options (PC, MP, and BCP insecticides at 1X, 1.5X, and 2X concentrations), not only was the decrease in the number of dividing cells as previously outlined identified, but also various types of chromosomal aberrations were scored and centralized in Table 4. In contrast to the insecticide treatments, no CA or nuclear abnormalities were observed during mitosis in NC grown in tap water. All prophase, metaphase, anaphase, and telophase stages were clearly visible without changes in the number or structure of chromosomes.

CAs, which include chromosomal bridges, laggard/vagrant chromosomes, chromosomal fragments, and decoiled chromosomes, indicated disturbances in the normal processes of cell division. Also, in the interphase, the presence of some cells with one or more micronuclei (MN) was observed.

Chromosome bridges and fragments are generally caused by the clastogenic action of the insecticides on DNA strands (structural aberrations), whereas lagging/vagrant chromosomes and chromosome losses represent the result of the aneugenic effects of these pesticides, which affect the number of chromosomes due to mitotic spindle abnormalities [61]. Chromosomal bridges are associated with further chromosome breakage, aneuploidy or polyploidy and sometimes even cell cycle arrest [62]. Vagrant and laggard chromosome presence generates, invariably, an unequal number of chromosomes in the resulting cells after division [63]. Acentric fragments, lacking the centromere, are unbalanced and distributed during cell division to daughter cells, some of them being lost in successive divisions. Depending on the number of genes contained, the loss of an acentric fragment may adversely affect the phenotype or may even be incompatible with the viability of the carrier organism [64].

Identifying the appearance of MNs is a simple and efficient way of testing the possible mutagenic effect of pesticides in general. They are visible in the interphase, and their presence and number represent definite proof of the genotoxic (clastogenic and aneugenic) effects of the tested insecticides [21]. MNs are the result of unrepaired or incorrectly repaired DNA damage, that is, of chromosomal aberrations (breaks and chromosome losses) during mitotic division [65].

At 24 h, all three insecticides induced chromosomal aberrations in a concentration-dependent manner. Interestingly, after 48 h of treatment, the number of chromosomal aberrations decreased, especially for MP treatment. One possible explanation for these observations might be that, after 24 h of treatment, the insecticides induced an adaptive response conferring the protection of root cells from genotoxic stress [66]. Also, it is possible that the cells with severely damaged DNA were directed to apoptosis, while in the surviving cells, DNA repair mechanisms operated [67].

As it was expected, positive control PC, which was a Deltamethrin containing insecticide, caused, at the 1X concentration, a significant number of chromosomal bridges and laggard chromosomes without inducing MN or other major abnormalities. At the 1.5X concentration, cells with MN and chromosomal fragments were observed, while at the highest concentration, this treatment induced a larger number of 1 MN cells but also cells with 3 MN (Figure 1).

BCP induced not only laggards and vagrant chromosomes but also a large number of cells with MN. In this case, the number of abnormal cells increased in correlation with the concentration used (Figure 2). The bio-chemical insecticide (BCP) investigated in this study belongs to the Naturalyte class of pesticides, a class that includes products derived from metabolites of living organisms. The main component of this insecticide is Spinosad, which is the fermentation product of the *Saccharopolyspora spinosa* aerobic bacteria. In recent years, several studies have been published indicating that Spinosad may have genotoxic effects on non-target species [68,69,70] or has been associated with various ecotoxicological events [71,72,73,74].

The results obtained in the present study are in agreement with those reported by Mendonca et. all, which showed that Spinosad exerted a mutagenic effect on *Tradescantia pallida* species at concentrations of 0.626, 1.25, and 2.5 g/mL [75], even though there is no total consensus among the data identified in the literature regarding Spinosad mutagenicity.

For example, in 1997, the U.S. Environmental Protection Agency stated that Spinosad has no mutagenic effect [76], results that were sustained by Yano et al. (who reported, in 2002, that Spinosad was no carcinogenic effect, at concentrations up to 0.05% to Fischer 344 rats) [77], Stebbins (who determined that chronic exposure of mice to Spinosad, i.e., 51 mg/kg/day for 18 months resulted in no tumor formation) [78] and Akmoutsou, which showed, using SMART test, that Spinosad does not have genotoxic activity on *D. melanogaster* [68]. On the other hand, Mansour and their team determined that Spinosad has genotoxic activity on rat bone marrow cells [79], while Asida et al., in a study from 2022, highlighted that a bio-pesticide containing Spinosad, had cytotogenotoxic effects (decrease in the mitotic index) on *A. cepa* genome [80]. The identified chromosomal aberrations were similar to those observed in the present study: sticky chromosomes, chromosome bridges, lagging chromosomes, and chromosome fragments [80]. Moreover, despite the fact that organic insecticides, such as BCP, are considered to be a safer alternative to chemical insecticides (such as those containing, e.g., Deltamethrin), there are studies showing that Spinosad, which is now registered for use in over 80 countries, poses a much greater risk to beneficial insects than previously thought, being toxic to pollinators [75]. In our study, the cytogenotoxic effects of the Spinosad-containing insecticide were mainly represented by laggard chromosomes and MN, the frequency of these aberrations being higher with the increase of insecticide concentration. BCP also determined the highest number of chromosomal aberrations among the analyzed insecticides, including PC.

The MP insecticide at the 1X concentration caused a high number of laggard/vagrant chromosomes and a few cells with 1 MN, and this effect is amplified at the 1.5X concentration. At 2X, the number of abnormalities slightly decreased (Figure 3). Thus, the most frequent cytogenetic abnormalities induced by the MP were the laggard/vagrant chromosomes, without a correlation between the insecticide concentration and the number of observed abnormalities. As mentioned above, MP’s stimulating effect might be due to the fact that it contains three different bacterial species (*B. thuringiensis*, *L. sphaericus* and *B. subtilis*) proven, in previous studies, to have bio-fertilizer potential by promoting plant growth [17,81,82].

In a very comprehensive review published in 2022, Oliveira-Filho and Grisolia showed that, in general, the *Bacillus* species commonly used in microbial insecticides have none to very low eco-toxicity and no cytogenotoxic effects on non-target organisms [83].

However, there are a number of studies showing that some Cry proteins (proteins that are produced by both *B. thuringiensis* and *L. sphaericus*) can induce MN formation in mice [61] and zebrafish erythrocytes [84]. Due to the fact that in our study, the MP insecticide induced the formation not only of MNs but also of other cytogenetic biomarkers (disturbed ana-telophase, chromosome laggards, stickiness, chromosomal bridges), it cannot be considered that the negative effects induced through the treatment with the MP insecticide could be due only to the substances with which the bacterial cells are associated in this commercial pesticide.

On the other hand, one of the main components of the MP insecticide is ‘*super concentrate vegetable extract*’, and even though there are no clear specifications of what is the precise chemical composition of this, usually a vegetable extract consists of a mixture of terpenes, flavonoids, tannins, saponins, alkaloids, and phenols [85]. In a review research on articles from 2010 to 2024 from different databases (ScienceDirect, PubMed, Web of Science, and Scopus), Ghanya et al. [85] summarized the data from 52 studies on plant extracts genotoxicity. The analyzed papers provided a brief overview of the extracts, the primary compounds identified, the plant parts used, the extraction method, the genotoxic tests, and the phytochemicals responsible for the genotoxicity effect. Among this, a total of six plant extracts showed no genotoxic effect, another 14 showed either genotoxic or mutagenic effect, and 14 showed anti-genotoxic effect against different genotoxic-induced agents. In addition, four plant extracts showed both genotoxic and non-genotoxic effects, and six plant extracts showed both genotoxic and antigenotoxic effects. What is very important to note is that in most of the articles, even though there were a number of suggestions regarding the compounds responsible for the genotoxic effects, none of the phytocompounds were individually tested for genotoxic potential to confirm the conclusions. Moreover, the mechanisms by which most plant extracts exert their genotoxic effect remain unidentified, and so more research is needed on the genotoxicity of plant extracts and their mechanisms of genotoxicity.

## 4. Conclusions

The risks of using pesticides to combat pests in agricultural crops were evaluated by applying the Allium bioassay, which proved effective in the assessment of the phyto- and cytogenotoxic effects of two insecticides recommended as relatively safe for humans and the environment. Tested insecticides are frequently used in agriculture, but, to date, there are no studies on their possible toxicity. However, there is a series of research on the clastogenic or phytostimulating effects of some of the components of the insecticides (e.g., analysis of the effect of spinosad, which is part of the composition of BCP or of the bacteria and plant extracts from MP constitution).

The tests conducted in this study showed that the microbial insecticide confers greater safety when applied even in higher doses than those recommended by the manufacturers compared with the biochemical insecticide, whose effects are similar to those induced by the chemical pesticide Deltamethrin. The obtained results suggest the clastogenic and aneugenic action of both insecticides, and, therefore, additional studies are needed to provide new information about the harmful potential of these pesticides both on the plant genome and on the environment. Moreover, the results support the idea that there is a need for prior testing of any type of pesticide before large-scale use, especially since the results of the Allium test are frequently similar to those of mammalian and human cell tests. The major problem related to insecticides is that of excessive use, without respecting the recommended concentrations and the break time between treatments.

## Figures and Tables

**Figure 1 jox-15-00035-f001:**
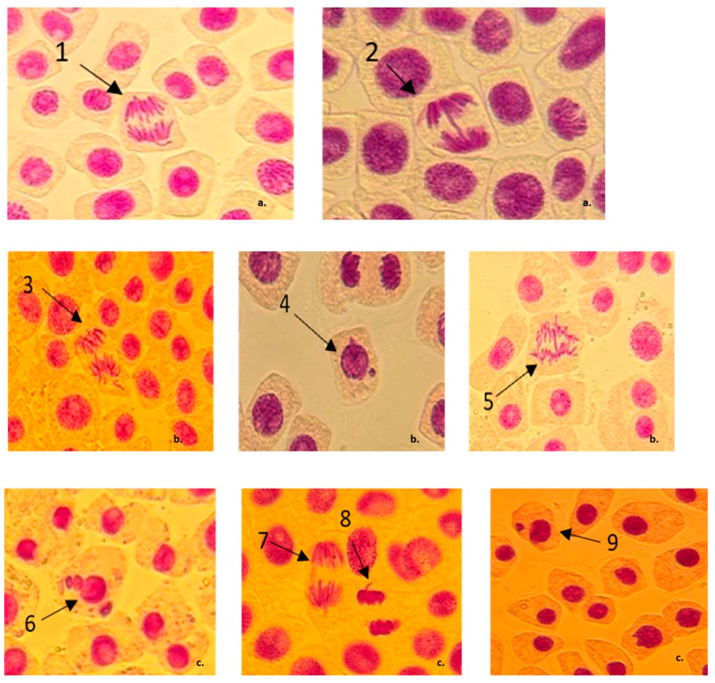
Chromosomal aberrations and nuclear irregularities observed in root tip cells of *A. cepa* after treatments with different concentrations of Deltamethrin-containing insecticide (PC). (**a**) MRFC concentration (1X variant): 1—anaphase with laggards and vagrant chromosomes; 2—anaphase bridge; (**b**) 1.5X concentration: 3—chromosome fragment/vagrant chromosome, 4—cell with 2 MNs, 5—anaphase bridge, (**c**) 2X concentration: 6—cell with 3 MNs, 7—laggards, 8—vagrant chromosomes, 9—cell with one MN.

**Figure 2 jox-15-00035-f002:**
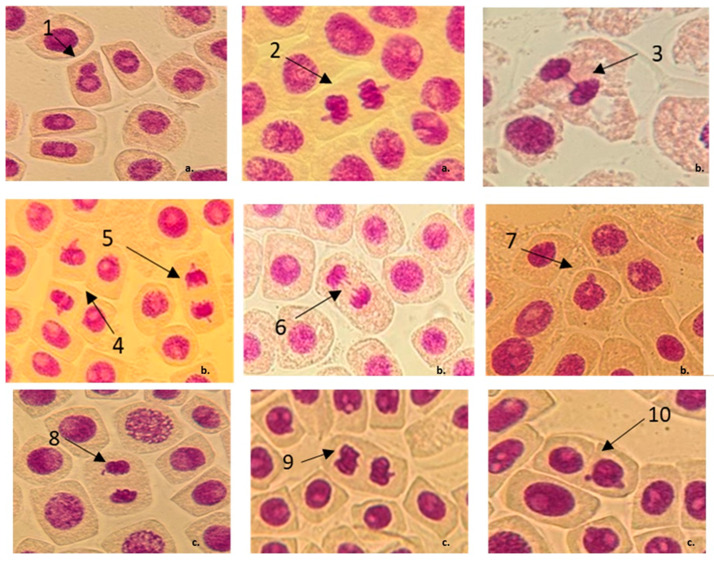
Chromosomal aberrations and nuclear irregularities observed in root tip cells of *A. cepa* after treatments with different concentrations of BCP insecticide. (**a**) MRFC concentration (1X concentration): 1—cell with 1 MN, 2—telophase with vagrant chromosomes, 3—telophase bridge; (**b**) 1.5X concentration: 4 and 5—telophase with vagrant chromosomes, 6—laggards, 7—cell with one MN; (**c**) 2X concentration: 8—chromosome fragment, 9—telophase with vagrant chromosomes, 10—cell with 2 MNs.

**Figure 3 jox-15-00035-f003:**
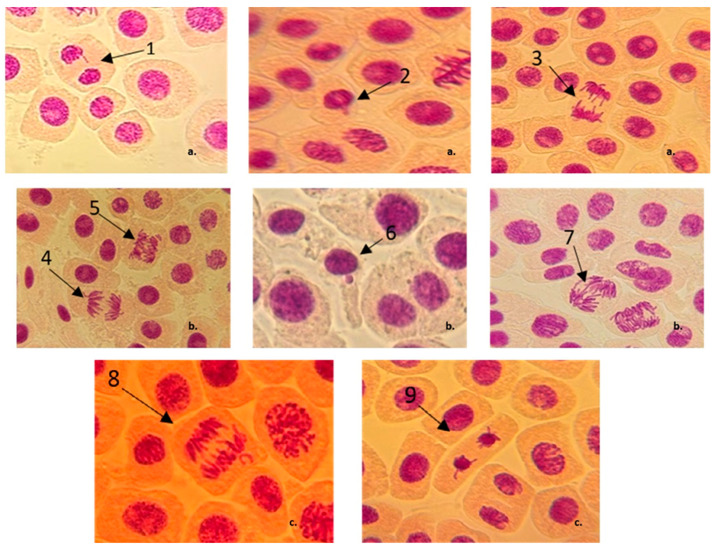
Chromosomal aberrations and nuclear irregularities observed in root tip cells of *A. cepa* following treatments with different concentrations of MP insecticide. (**a**) MRFC concentration (1X concentration): 1—telophase with vagrant chromosome, 2—cell with 1 MN, 3—anaphase with laggards and vagrant chromosomes; (**b**) 1.5X concentration: 4—anaphase with vagrant chromosomes, 5—anaphase with abnormal segregation, 6—cell with 1 MN, 7—anaphase bridge; (**c**) 2X concentration: 8—chromosomal fragment/vagrant chromosome, 9—telophase with laggards and vagrant chromosomes.

**Table 1 jox-15-00035-t001:** The percentage concentration of insecticide solutions in the two experimental variants and in the positive control.

Experimental Variant	Insecticide Concentration (%)		
	**PC**	**BCP**	**MP**
**MRFC (1X)**	0.031	0.04	2
**1.5 MRFC (1.5X)**	0.046	0.06	3
**2 MRFC (2X)**	0.062	0.08	4

MRFC = maximum recommended field concentration, 1.5X = 1.5-fold higher than MRFC, 2X = twofold higher than MRFC, PC = positive control (insecticide with Deltamethrin), BCP = bio-chemical insecticide, MP = microbial insecticide.

**Table 2 jox-15-00035-t002:** Comparative root length, root number, and root growth inhibition of *A. cepa* exposed to various concentrations of insecticides after 24 and 48 h.

Treatment			Root Length (cm) (Mean ± SD)	No. of Roots (Mean ± SD)	Root Growth Inhibition (%)
**NC**		**24 h**	2.2 ± 1.8	30.5 ± 2.04	-
		**48 h**	2.75 ± 1.5	37 ± 1	-
**PC**	**1X**	**24 h**	1.025 ± 0.6	46 ± 1.6	53.41
		**48 h**	1.075 ± 0.9	30.5 ± 1.5	60.91
	**1.5X**	**24 h**	0.675 ± 0.4	42 ± 0.8	69.32
		**48 h**	0.8 ± 0.6	35 ± 7.7	70.91
	**2X**	**24 h**	0.625 ± 0.3	45.5 ± 2.85	71.59
		**48 h**	0.55 ± 0.54	31 ± 4	80.00
**BCP insecticide**	**1X**	**24 h**	1.52 ± 0.1	41 ± 1.4	30.91
		**48 h**	0.75 ± 0.1	28.5 ± 4.9	72.73
	**1.5X**	**24 h**	0.93 ± 0.5	20 ± 0.1	57.73
		**48 h**	0.82 ± 0.2	19.5 ± 9.1	70.18
	**2X**	**24 h**	0.68 ± 0.2	27.5 ± 3.5	69.09
		**48 h**	0.57 ± 0.2	13.5 ± 0.7	79.27
**MP insecticide**	**1X**	**24 h**	1.85 ± 0.7	30.5 ± 9.1	15.91
		**48 h**	2.45 ± 0.7	34 ± 7.07	10.91
	**1.5X**	**24 h**	1.7 ± 0.6	29 ± 8.4	22.73
		**48 h**	2.42 ± 0.5	33 ± 1.6	12.00
	**2X**	**24 h**	2.37 ± 0.4	33.5 ± 0.7	**−4.45**
		**48 h**	3.3 ± 0.2	37.5 ± 0.1	**−20**

1X = MRFC = maximum recommended field concentration, 1.5X = 1.5-fold higher than MRFC, 2X = twofold higher than MRFC, SD = standard deviation, NC = negative control (distilled water), PC = positive control (insecticide with Deltamethrin), BCP = bio-chemical insecticide, MP = microbial insecticide.

**Table 3 jox-15-00035-t003:** Cytogenetic analysis of *A. cepa* root tips exposed to insecticides in correlation with the concentration and time of exposure.

Treatment			TNDC ± SD	MI	Pro. (%)	Meta. (%)	Ana. (%)	Telo. (%)
**NC**		**24 h**	270 ± 8.9	33.75	39.62	29.97	11.4	21.85
		**48 h**	362 ± 1.5	45.25	34.25	32.9	13.5	26.6
**PC**	**1X**	**24 h**	241 ± 6.2	30.13	36.51	19.08	18.6	25.7
		**48 h**	**168 ± 4.2 ^a^**	**21.13 ^a^**	**26.62 ^a^**	**30.76 ^a^**	**21.3 ^a^**	**21.3 ^a^**
	**1.5X**	**24 h**	**184 ± 8.4 ^b^**	**23 ^b^**	**26.1 ^b^**	**36.4 ^b^**	**17.4 ^b^**	**20.1 ^b^**
		**48 h**	**227± 4.4 ^b^**	**28.34 ^b^**	**24.66 ^b^**	**40.08 ^b^**	**16.4 ^b^**	**18.5 ^b^**
	**2X**	**24 h**	**20 ± 4.3 ^a^**	**2.5 ^a^**	**15 ^a^**	**55 ^a^**	**15 ^a^**	**15 ^a^**
		**48 h**	214 ± 9.2	26.75	28.5	36.4	14.5	20.6
**BCP**	**1X**	**24 h**	200 ± 7.5	25	33	21	17	29
		**48 h**	**130 ± 6.25 ^b^**	**16.25 ^b^**	**37.7 ^b^**	**25.4 ^b^**	**12.3 ^b^**	**24.6 ^b^**
	**1.5X**	**24 h**	**166 ± 3.6 ^b^**	**20.75 ^b^**	**36.1 ^b^**	**16.9 ^b^**	**17.5 ^b^**	**29.5 ^b^**
		**48 h**	**165 ± 3.6 ^b^**	**20.62 ^b^**	**34.5 ^b^**	**22.4 ^b^**	**17.6 ^b^**	**25.5 ^b^**
	**2X**	**24 h**	**151 ± 1.9 ^b^**	**18.8 ^b^**	**29.1 ^b^**	**25.8 ^b^**	**19.9 ^b^**	**25.2 ^b^**
		**48 h**	**171 ± 7.8 ^b^**	**21.38 ^b^**	**36.8 ^b^**	**13.7 ^b^**	**16.9 ^b^**	**27.5 ^b^**
**MP**	**1X**	**24 h**	253 ± 6.2	31.62	41.89	18.97	14.62	24.5
		**48 h**	287 ± 6.9	35.87	31.35	29.96	20.55	18.11
	**1.5X**	**24 h**	**198 ± 6.2 ^b^**	**24.75 ^b^**	**35.87 ^b^**	**22.72 ^b^**	**16.66 ^b^**	**22.72 ^b^**
		**48 h**	**220 ± 4.4 ^a^**	**27.5 ^a^**	**37.72 ^a^**	**22.72 ^a^**	**15.45 ^a^**	**24.09 ^a^**
	**2X**	**24 h**	219 ± 2.1	27.37	34.7	22.83	17.8	24.65
		**48 h**	221 ± 7.3	27.62	35.75	26.69	18.55	19.1

1X, 1.5X, 2X = experimental variants, TNDC = total number of cells in the division, SD = standard deviation, MI = mitotic index, Pro. % = % of prophases out of the total number of cells in the division, Meta. % = % of metaphases from the total number of cells in division, Ana. % = % of anaphases from the total number of cells in the division, Telo. % = % telophases out of total dividing cells, ^a^ = statistically significant values, *p* ≤ 0.05 compared with a negative control with *t*-test and Bonferroni Correction, ^b^ = highly statistically significant values, *p* ≤ 0.01 compared with a negative control with *t*-test and Bonferroni Correction.

**Table 4 jox-15-00035-t004:** Chromosome aberrations on root meristematic cells of *A. cepa* after treatment with different concentrations of insecticides.

	PC						BCP						MP					
	1X		1.5X		2X		1X		1.5X		2X		1X		1.5X		2X	
	24 h	48 h	24 h	48 h	24 h	48 h	24 h	48 h	24 h	48 h	24 h	48 h	24 h	48 h	24 h	48 h	24 h	48 h
**Chromosomal bridges**	2	0	0	4	0	3	0	0	0	1	1	0	0	1	1	1	0	0
**Vagrant/laggard**	8	6	4	11	0	5	5	0	8	5	4	11	12	0	7	13	14	2
**Other modifications ***	0	0	1	2	0	0	1	0	0	0	0	0	1	0	1	0	0	0
**TCA**	**10**	**6**	**5**	**17**	**0**	**8**	**6**	**0**	**8**	**6**	**5**	**11**	**13**	**1**	**9**	**14**	**14**	**2**
**Cells with 1 MN**	0	0	0	1	23	1	5	8	6	10	4	29	3	0	2	1	0	0
**Cells with 2-more MNs ****	0	0	0	1	6	0	3	0	0	1	0	4	0	0	0	0	0	0
**TCNM**	**0**	**0**	**0**	**2**	**29**	**1**	**8**	**8**	**6**	**11**	**4**	**33**	**3**	**0**	**2**	**1**	**0**	**0**

* = chromosomal fragment and/or decoiled chromosomes; ** = we have found cells with 3 and 6 MNs; TCA = total number of chromosomal aberrations; TCNM = total number of cells with MN; 1X, 1.5X, 2X = experimental variants; PC = positive control (insecticide with Deltamethrin); BCP = bio-chemical insecticide; MP = microbial insecticide.

## Data Availability

The original contributions presented in this study are included in the article. Further inquiries can be directed to the corresponding author.

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
