# Peer review of "Evaluation of Clastogenic and Aneugenic Action of Two Bio-Insecticides Using Allium Bioassay"

_jox, 2025, doi:10.3390/jox15020035_

Round 1
Reviewer 1 Report
Comments and Suggestions for Authors
This research investigates the phyto- and cyto-genotoxic effects of two bio-pesticides on Allium cepa, using various concentrations to assess potential environmental and human health risks. The study highlights the clastogenic and aneugenic actions of the tested insecticides, suggesting that prior testing is essential before large-scale use. However, the manuscript requires extensive revision before it can proceed to the next stage for acceptance.
1. The methods section should be more detailed regarding the preparation of insecticide solutions. Specifically, elaborate on how the insecticides were diluted and any controls used to avoid potential variations. This will ensure reproducibility.
2. The statistical analysis section requires further clarification. A more robust explanation of the chosen statistical methods, especially regarding the use of T-tests and Bonferroni correction, would enhance the credibility of the findings.
3. The study mentions a negative control using distilled water and a positive control (Deltamethrin). Please justify the choice of Deltamethrin as a positive control in greater detail. What criteria were used to select this particular chemical, and how does it compare with the bio-pesticides tested?
4. The paper should mention the sample size more explicitly. Replicates per treatment group (root tips, number of cells analyzed) should be specified in greater detail to enhance the robustness of the data.
5. The discussion of the mechanisms behind observed genotoxicity (clastogenic and aneugenic effects) could benefit from a more thorough explanation. Are these effects consistent with the known modes of action of the pesticides tested?
6. The study utilizes various concentrations of insecticides. However, it would be beneficial to explain how these concentrations were chosen, especially the higher concentrations (1.5X and 2X MRFC), in the context of their relevance to real-world application.
7. The paper claims that results from Allium cepa can be extrapolated to human health. More rigorous justification of this extrapolation is needed, including a discussion of potential limitations and differences between plant and human cell systems.
8. The results section presents statistical significance but lacks depth in interpreting what these results imply. Further discussion of the biological relevance of the observed clastogenic and aneugenic effects is needed.
9. The tables and figures, while informative, could benefit from clearer labelling and consistency in the presentation of standard deviations and other metrics. Consider including clearer captions that define abbreviations and explain the experimental setup more thoroughly.
10. The composition of the microbial pesticide (MP) is mentioned, but more detail is needed on the individual contributions of the various bacterial species and other components. What role do these play in the observed results?
Minor Comments:
· There are minor typographical errors throughout the paper (e.g., "phyto- and cyto-genotoxicity" vs. "phyto- and cytogenotoxicity"). A thorough proofread would enhance the overall readability.
· Ensure consistency in terminology, especially in terms like "bio-pesticides" vs. "bio-insecticides." The use of one term throughout the manuscript will improve clarity.
· Some references appear to be incomplete or inconsistent in format. Ensure all references conform to the journal's required style guide.
· Ensure that figures and tables referenced in the text are clearly numbered and correspond appropriately. Some references to figures (e.g., Figure 1) seem to be incorrectly cited.
· The results regarding the microbial insecticide (MP) showing phytostimulatory effects at the highest concentration (2X MRFC) should be discussed in more detail. Are these effects potentially beneficial or harmful?
Author Response
- The methods section should be more detailed regarding the preparation of insecticide solutions. Specifically, elaborate on how the insecticides were diluted and any controls used to avoid potential variations. This will ensure reproducibility.
answer: problem address in lines 148-154
- The statistical analysis section requires further clarification. A more robust explanation of the chosen statistical methods, especially regarding the use of T-tests and Bonferroni correction, would enhance the credibility of the findings.
answer: we supplemented the expalnation regarding Bonferroni correction (lines 197-201).
- The study mentions a negative control using distilled water and a positive control (Deltamethrin). Please justify the choice of Deltamethrin as a positive control in greater detail. What criteria were used to select this particular chemical, and how does it compare with the bio-pesticides tested?
answer: we added more refrecnces regarding the phytotoxic and genotoxic effect of Delthametrin (lines 103-109)
- The paper should mention the sample size more explicitly. Replicates per treatment group (root tips, number of cells analyzed) should be specified in greater detail to enhance the robustness of the data.
answer: as we mentioned, we used three replicates/experimental variant /tipe of solution (NC, PC, BCP, MP/time interval and for cytogenotoxicity we used 6 root tips. We analyez 500c cells/slide (lines 175-180).
- The discussion of the mechanisms behind observed genotoxicity (clastogenic and aneugenic effects) could benefit from a more thorough explanation. Are these effects consistent with the known modes of action of the pesticides tested?
answer: Until now, there is no data regarding the mode of action of these pesticides. There is data only regarding some of the compounds that are part of the composition of these pesticides (for example, spinosad, which is part of BCP, or the different bacterial strains in the composition of MP). More imformation regardin the clastrogenic and aneugenic effect of this compounds were detailed further (ex. lines 239-251, lines 277-280, lines 315-320, lines 339-348, 358-374).
- The study utilizes various concentrations of insecticides. However, it would be beneficial to explain how these concentrations were chosen, especially the higher concentrations (1.5X and 2X MRFC), in the context of their relevance to real-world application.
answer: see lines 208-211, 431-432
- 7. The paper claims that results from Allium cepa can be extrapolated to human health. More rigorous justification of this extrapolation is needed, including a discussion of potential limitations and differences between plant and human cell systems.
answer: we have rephrased the statement both in abstract and in text (”since the results of the A. cepa tests showed high sensitivity and good correlation when compared with other test systems, e.g. mammals”) and give more details (lines 92-93).
- The results section presents statistical significance but lacks depth in interpreting what these results imply. Further discussion of the biological relevance of the observed clastogenic and aneugenic effects is needed.
answer: we addressed that
- The tables and figures, while informative, could benefit from clearer labelling and consistency in the presentation of standard deviations and other metrics. Consider including clearer captions that define abbreviations and explain the experimental setup more thoroughly.
answer: we addressed that
- The composition of the microbial pesticide (MP) is mentioned, but more detail is needed on the individual contributions of the various bacterial species and other components. What role do these play in the observed results?
answer: taking into account that it is the first study that analyzes the phytotoxic and cytogenotoxic effect of this insecticide, we cannot provide a more detailed analysis regarding its mode of action. As mentioned previously, the attempt to explain our observations is based on data from the literature that refer to individual components (lines 375-414).
- The results regarding the microbial insecticide (MP) showing phytostimulatory effects at the highest concentration (2X MRFC) should be discussed in more detail. Are these effects potentially beneficial or harmful?
answer: we cannot formulate a clear conclusion to this observation because, as it appears from the data obtained, even at 2X concentrations the total number of chromosomal aberrations was high. That is why it was mentioned that further investigations are needed to determine whether the phytostimulatory effect on root growth is, in total, beneficial or not
Reviewer 2 Report
Comments and Suggestions for Authors
The study presents a clear and well-structured experimental design with intriguing results. These results would be more effectively highlighted using graphs rather than tables, as graphs would allow the reader to more easily visualize and understand the differences observed between the two compounds and the PC. Additionally, they would clarify how the MP appears to be safer than the BMP, which shows overlapping effects with the PC in some cases.
Below are just a few minor revisions that should be made:
Line 57-62: repetition of the same concept
Line 96-108: partial repetition of the same concept. Unify the concept you want to express.
Line 132-138: Format the line numbering table 1 and standardize the units of measurement, either in % or all in g/L.
Line 165-166: Standardize the formula indicated for the calculation of MI with the others previously entered in lines 152-154.
Line 219: The results indicated refer to Table 3, not Table 2.
Line 230: Insert the table on a single page
Line 261: Insert the acronym MN first in the text when referring to micronuclei for the first time
Line 272-274: Is there a bibliographic source to refer to that confirms this hypothesis? An additional analysis that could be performed to confirm the apoptosis mechanism is the acridine orange staining.
Line 288-291: Please insert bibliographic references that support this concept, as it is crucial to understand that it is not the MP causing negative effects, but rather the substances with which the bacterial cells are associated in this commercial pesticide.
Additionally, it is important to specify:
1. The decomposition time to understand how long it persists in the environment and whether there are potential risks to animals and humans.
2. Specificity: Considering that, as stated, the growth hormone inhibitor for insects could be responsible for disrupting cell division in plants, it would be advisable to indicate the pathway through which this mechanism occurs. This is important, as the definition of a bio-pesticide also implies a high specificity toward target pests. Is this pathway present in other animals or in humans?

Author Response
- Line 272-274: Is there a bibliographic source to refer to that confirms this hypothesis? An additional analysis that could be performed to confirm the apoptosis mechanism is the acridine orange staining.
answer: we addressed this, in the lines 315-320. Thank you for you advice. Since we plan to continue the research, we will certainly apply this test as well.
- Line 288-291: Please insert bibliographic references that support this concept, as it is crucial to understand that it is not the MP causing negative effects, but rather the substances with which the bacterial cells are associated in this commercial pesticide.
answer: we addressed this, see lines 375-414
- The decomposition time to understand how long it persists in the environment and whether there are potential risks to animals and humans.
answer: because these insecticides are relatively new, we have not identified any articles that analyze the time and manner of their decomposition. the potential harmful effect on the ecosystem of animals and humans could possibly be assessed on the basis of individual components alone, but we do not think that would be correct. That is why I mentioned that further research is needed.
- Specificity: Considering that, as stated, the growth hormone inhibitor for insects could be responsible for disrupting cell division in plants, it would be advisable to indicate the pathway through which this mechanism occurs. This is important, as the definition of a bio-pesticide also implies a high specificity toward target pests. Is this pathway present in other animals or in humans?
answer: we think that we addressed this, even if we rephrased the concept (line 375-414)
Round 2
Reviewer 1 Report
Comments and Suggestions for Authors
The manuscript has been thoroughly revised and is now suitable for publication.